# Substitution of arginine 219 by glycine compromises stability, dimerization, and catalytic activity in a G6PD mutant

Omar Zgheib [1✉], Kamonwan Chamchoy[2], Thierry Nouspikel[1], Jean-Louis Blouin[1], Laurent Cimasoni[3], Lina Quteineh[1] & Usa Boonyuen [4✉]

Glucose-6-phosphate dehydrogenase (G6PD) deficiency is one of the most common enzymopathies in humans, present in approximately half a billion people worldwide. More than 230 clinically relevant G6PD mutations of different classes have been reported to date. We hereby describe a patient with chronic hemolysis who presents a substitution of arginine by glycine at position 219 in G6PD protein. The variant was never described in an original publication or characterized on a molecular level. In the present study, we provide structural and biochemical evidence for the molecular basis of its pathogenicity. When compared to the wild-type enzyme, the Arg219Gly mutation markedly reduces the catalytic activity by 50-fold while having a negligible effect on substrate binding affinity. The mutation preserves secondary protein structure, but greatly decreases stability at higher temperatures and to trypsin digestion. Size exclusion chromatography elution profiles show monomeric and dimeric forms for the mutant, but only the latter for the wild-type form, suggesting a critical role of arginine 219 in G6PD dimer formation. Our findings have implications in the development of small molecule activators, with the goal of rescuing the phenotype observed in this and possibly other related mutants.

[1] Division of Genetic Medicine, Department of Diagnostics, Geneva University Hospitals, Geneva, Switzerland. [2] Princess Srisavangavadhana College of Medicine, Chulabhorn Royal Academy, Bangkok, Thailand. [3] Division of Pediatric Haematology, Department of Pediatrics, Geneva University Hospitals, Geneva, Switzerland. [4] Department of Molecular Tropical Medicine and Genetics, Faculty of Tropical Medicine, Mahidol University, Bangkok, Thailand. ✉email: omar.zgheib@hcuge.ch; usa.boo@mahidol.edu

Glucose-6-phosphate dehydrogenase (G6PD) deficiency is one of the most common enzymopathies in humans, present in approximately half a billion people worldwide, with a predominance in sub-Saharan Africa, the Arabian Peninsula, and South Asia, where it confers a survival advantage to *Plasmodium falciparum* infections[1]. G6PD functions to protect red blood cells from oxidative damage by generating a reduced form of nicotinamide adenine dinucleotide phosphate (NADPH) via the pentose phosphate pathway. As the sole cellular source of NADPH in red blood cells, G6PD is therefore crucial for maintaining an appropriate level of glutathione, thus shielding against oxidative damage and preventing hemolytic anemia.

G6PD deficiency is an X-linked genetic disorder. The gene is located on Xq28, a well-known hotspot at the telomeric region of the X chromosome q arm. While homozygosity is rare in females, heterozygosity is associated with a variable phenotype due to X-inactivation. Males have only one copy of the gene. To date, more than 230 clinically relevant G6PD mutations have been reported[1].

A classification of variants, established in 1971, in use until 2022, divides G6PD variants into five classes, the first three of which are pathogenic[2,3]. Class I variants are those observed in chronic nonspherocytic hemolytic anemia (CNSHA) in the absence of infections, oxidant drugs, or ingestion of fava beans. Class II and III variants have less than 10% and 10 to 60% residual activity, respectively. Normal activity is observed in Class IV variants, while Class V variants show increased activity. The World Health Organization updated this classification in 2022, which now includes the following four classes, with activity and phenotype shown in parentheses: A (<20% activity; CNSHA), B (<45% activity; acute, triggered anemia), C (60–150% activity; no hemolysis), and U (uncertain clinical significance, regardless of activity)[4].

We hereby describe a G6PD variant in a 15-year-old man known for G6PD deficiency with undetectable activity, corresponding, therefore, to a Class A (previously Class I) CNSHA variant. Further, we provide structural, biophysical and biochemical evidence for the molecular basis of its pathogenicity.

The patient was known for chronic hemolytic anemia and presented an episode of hemolysis at age 14 in the context of a viral influenza A infection. His main symptoms included fatigue and weight and appetite loss. Hemolysis, pancytopenia, and an inflammatory syndrome were observed in his laboratory workup. Total and conjugated bilirubin levels were at 46.5 μmol/L and 18.3 μmol/L (N 0–10, 0–5.2), and improved 2 weeks later (total bilirubin at 19 μmol/L). Six months after resolution of the viral episode, macrocytic anemia persisted, with hemoglobin at 120 g/l (N 130–160), erythrocyte mean corpuscular volume at 103.7 fl (N 78–98), and reticulocytes at 41.4 ‰ (N 5–15). Abdominal ultrasound was unremarkable. Interestingly, initial G6PD activity measured at an external laboratory showed 15 to 20% activity; subsequent dosage at the national reference laboratory (Zurich, Switzerland) showed null activity. The result from the latter measurement was then retained, with clinical diagnosis of Class A G6PD deficiency. At this point, our medical genetics team was solicited for a molecular diagnosis in order to confirm the variant's pathogenicity, given the discordant activity data from the two laboratories.

The patient's mother was known for reticulocytosis and macrocytosis; and his maternal grandmother was known for reticulocytosis and macrocytic anemia. His maternal uncle was known for G6PD deficiency, based on enzymatic activity measurement, with no genetic testing done. Both mother and uncle had presented neonatal jaundice, interestingly more severe in his mother's case. The rest of the family history was noncontributory.

The patient was found to carry, at the hemizygous state, a missense variant in the *G6PD* gene (NM_000402):c.745 A > G:p.(Arg249Gly) in exon 7, corresponding to an arginine-to-glycine amino acid change at position 219 in the canonical transcript (A655G base change). Segregation analysis showed that he inherited the variant from his mother, consistent with an X-linked recessive mode of inheritance. The variant was also found in his maternal uncle and maternal grandmother. The uncle was hemizygous, and the mother and grandmother heterozygous, as expected.

The variant was absent from the Genome Aggregation Database and public archives that report evidence-based relationships among human variations and phenotypes, namely Human Gene Mutation Database, Leiden Open Variation Database, and ClinVar, a database hosted by the National Center for Biotechnology Information. The amino acid change was predicted as pathogenic by all used algorithms (SIFT, PolyPhen, MutationTaster and CADD).

The same variant, named G6PD Meyer, was cited in a recent review on G6PD, but never described in an original publication[3]. We therefore sought to extensively characterize it on a molecular level in the present study. We purified bacterially expressed protein and performed circular dichroism spectroscopy to explore secondary structure preservation. Biochemical characterization was done to determine how the mutation affects enzyme catalytic activity. We then carried out thermal stability assays as well as assays for susceptibility to chemical denaturation and trypsin digestion. Finally, we performed size exclusion chromatography to study the effect of the mutation on the oligomeric state.

## Results

**Structural prediction**. Based on G6PD structure, Arg219 is predicted to play an important role in dimer stabilization and is conserved across species, from *C. elegans* to humans. Nitrogen epsilon (ε) and nitrogen eta 1 (η1) interact with the backbone carboxyl oxygens of His374 and Asp375. N ε is also proximal to the Asn229 sidechain. Nitrogen eta 2 (η2) interacts with the backbone carboxyl oxygen of Asp228 and Asn229, as well as the side chain atoms of Asn229 (Fig. 1A, B).

**Protein expression and purification**. G6PD proteins were successfully expressed in bacteria and purified to homogeneity by affinity chromatography. Sodium dodecyl-sulfate polyacrylamide gel electrophoresis analysis of the recombinant proteins is shown in Supplementary Fig. 1, with protein purity greater than 95%. It has been previously shown that His-tagged G6PD protein has no effect on catalytic activity, structure or stability[5]; therefore, all recombinant G6PD proteins in this study were His-tagged and characterized without removing the tag.

**Biochemical characterization of G6PD Meyer**. Biochemical characterization was carried out to determine the effects of the Arg219Gly mutation on G6PD catalytic activity (Table 1). This was 50-fold lower ($k_{cat}$ 6 s$^{-1}$) compared to the wild-type (WT) enzyme ($k_{cat}$ 326 s$^{-1}$). Affinity toward NADP$^+$ binding was only slightly affected in the Arg219Gly variant ($K_m$ 18.1 ± 2.7 μM) compared to WT ($K_m$ 12.9 ± 2.9 μM). Binding affinity toward glucose-6-phosphate (G6P) substrate was not affected by the Arg219Gly mutation, with $K_m$ values of 49.5 ± 6.5 and 46.1 ± 3.8 μM for Arg219Gly and WT, respectively. Since $K_m$ reflects the affinity toward substrate binding, this is not surprising, given that the Arg219Gly mutation is not part of the catalytic site and thus is not expected to impact substrate binding but rather dimer formation, as predicted and shown later. Thus, the Arg219Gly mutation only has a minor effect on binding affinity toward both substrates while markedly decreases the catalytic efficiency of the enzyme.

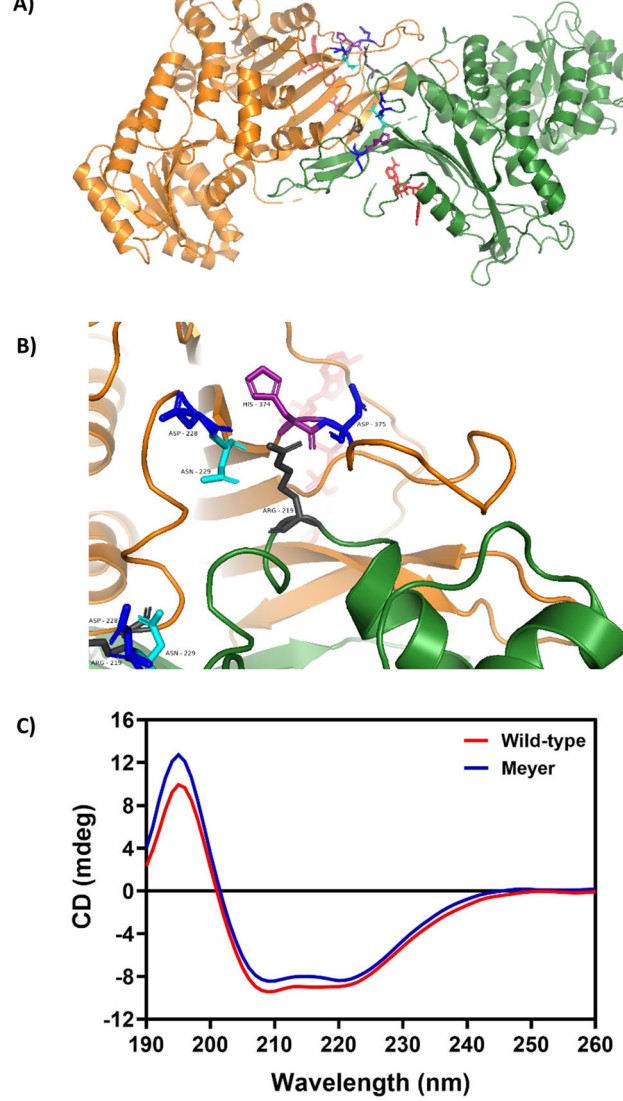

**Fig. 1 G6PD dimer structure showing Arg219 interactions and circular dichroism spectra of recombinant wild-type and Arg219Gly G6PD.**
**A** PyMOL illustration of wild-type G6PD dimer structure adapted from the dimer of dimers structure[7], showing Arg219 in gray and relevant interacting residues at the dimer interface: Asp228 and 375 in blue, Asn229 in cyan, and His 374 in purple. Structural NAPD$^+$ is shown in red. **B** Close-up view of the dimer. Arg219 (center) interacts via its N ε and N η1 (right) with the backbone carboxyl O of His374 and Asp375. N ε is also proximal to the Asn229 sidechain. Arg N η2 (left) interacts with the backbone carboxyl O of Asp228 and Asn229, as well as the side chain atoms of Asn229. **C** Far ultraviolet CD spectra of recombinant human G6PD wild-type and Arg219Gly mutant. The protein concentration was 0.15 mg/ml and CD spectra were recorded between 190 and 260 nm at a scan rate of 50 nm/min using a Jasco spectrometer, model J-815.

**Table 1 Steady-state kinetic parameters of recombinant G6PD enzymes.**

| Construct | Amino acid change | $k_{cat}$ (s$^{-1}$) | $K_m$NADP$^+$ (μM) | $K_m$G6P (μM) |
|---|---|---|---|---|
| G6PD WT | - | 326 | 12.9 ± 2.9 | 46.1 ± 3.8 |
| G6PD Meyer | Arg219Gly | 6 | 18.1 ± 2.7 | 49.5 ± 6.5 |

Experiments were performed in triplicate, and data are presented as mean ± SD.

**Secondary structure analysis.** The secondary structure of G6PD protein was determined by circular dichroism (CD) to monitor conformational changes caused by the mutation. Based on CD spectra, the Arg219Gly mutation shared a similar absorption pattern to that of the WT protein, showing two negative peaks at 208 and 222 nm, which are characteristic of α-helical proteins (Fig. 1C). This indicates that the Arg219Gly mutation has no effect on the secondary structure of the protein. A slight difference in signal intensity between WT and mutant proteins could be caused by a small difference in protein concentration.

**Structural stability analysis of G6PD Meyer.** G6PD mutations have been shown to be linked with G6PD deficiency by causing structural destabilization. Structural stability alterations caused by the Meyer mutation were thoroughly investigated. Heat treatment causes protein unfolding and loss of tertiary/quaternary structure, and could be used to assess protein structural stability. Fluorescence-based thermal shift assays revealed that the $T_m$ of the mutant (47.04 °C) was ∼6 °C lower than the WT enzyme (52.99 °C) in the absence of NADP$^+$ (Fig. 2A, B). The presence of NADP$^+$ was found to stabilize the protein, increasing the $T_m$ values for both mutant and WT enzymes. In the presence of 10 and 100 μM of NADP$^+$, $T_m$ values for the Arg219Gly variant were 51.61 and 54.02 °C, respectively. The mutant enzyme was found to be more susceptible to chemical denaturation and trypsin digestion than the WT enzyme, indicating its structural instability. The WT enzyme was structurally more stable than the Meyer mutation, with greater resistance to trypsin digestion, retaining higher enzyme activity than the mutant enzyme both in the absence and presence of NADP$^+$ (Fig. 2C, D). In the absence of NADP$^+$, the residual enzyme activity of WT was 6%, whereas the mutant retained only 1.5% of its activity. While NADP$^+$ greatly enhanced structural stability in the WT enzyme, it only slightly improved structural stability in the mutant. Similarly, the WT was more resistant to chemical denaturation than the mutant enzyme, losing half of its activity at 0.22 M guanidine hydrochloride (Gdn-HCl) compared to 0.1 M for the mutant (Fig. 2E). Furthermore, thermal inactivation assays also indicated that the Arg219Gly variant was functionally deficient as a result of structural instability. The mutant enzyme lost half of its activity at 44.48 °C, whereas the WT enzyme lost half of its activity at 50.16 °C (Fig. 2F). Hence, stability analysis indicated that the Arg219Gly mutation caused protein structural destabilization contributing to enzyme deficiency.

**Size exclusion chromatography of G6PD Meyer.** Because Arg219 is located at the dimer interface, replacing it with glycine could have a remarkable impact on dimerization, causing dimer to monomer dissociation and decreased enzyme activity. Size exclusion chromatography was then carried out to determine the oligomeric state of the Arg219Gly mutant (Fig. 3). While the WT enzyme predominated as a dimer, the mutant showed two major populations on the chromatogram, corresponding to dimeric and monomeric forms. Size exclusion chromatography thus confirmed that arginine substitution by glycine at residue 219 resulted in dimer to monomer dissociation.

**Discussion**
G6PD exists in its active form as a tetramer or dimer, depending on pH conditions. The monomer is 515 residues long and essentially composed of two domains; an N-terminal domain (residues 27–200) that includes the catalytic NADP$^+$ binding site (residues 38–44) and a second domain that comprises nine anti-parallel β sheets and includes the dimerization interface, adjacent to the structural NADP$^+$ binding site. It is interesting to note that

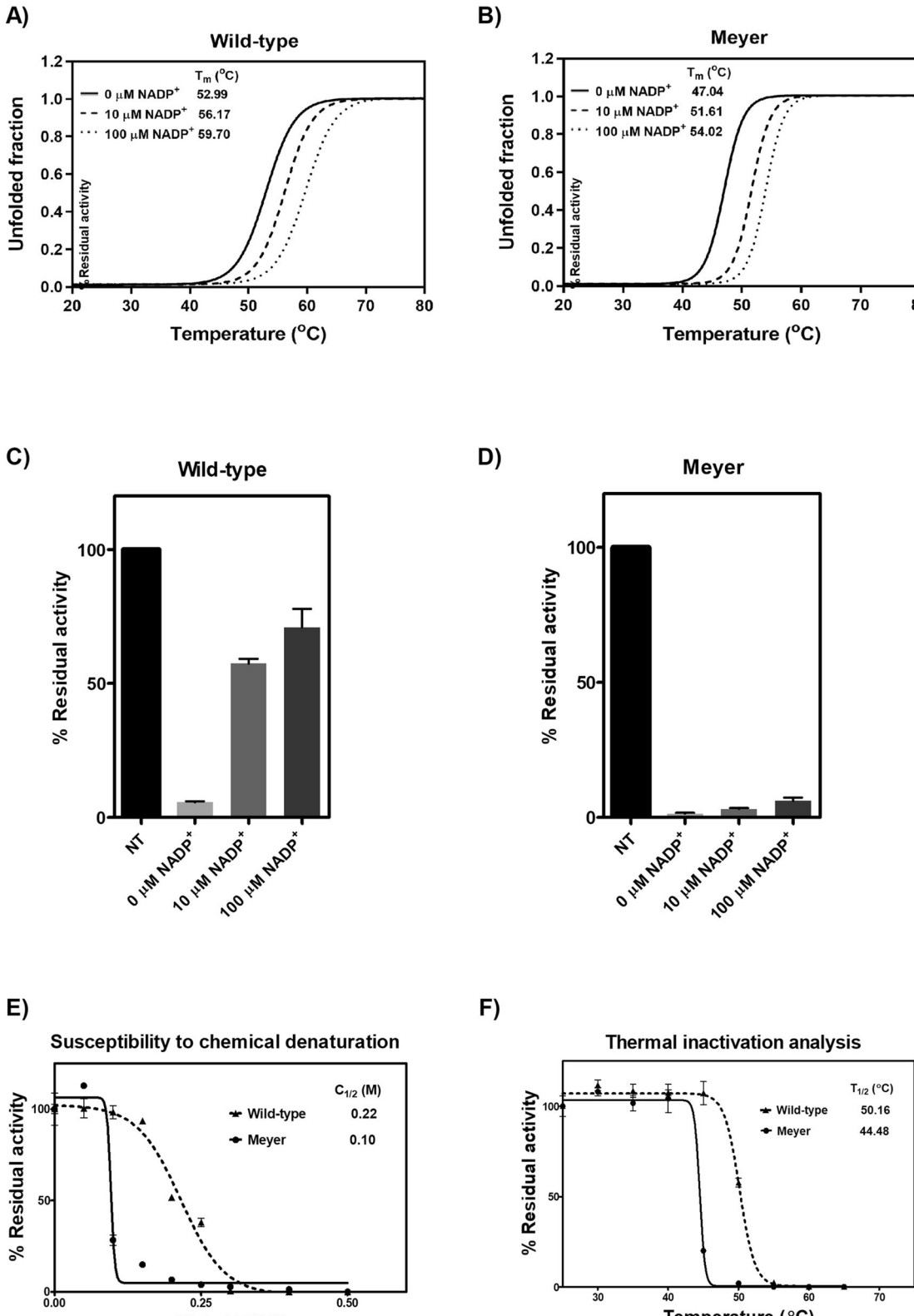

G6PD is the only known enzyme containing a second 'structural' NADP+ site in addition to the catalytic site. The structural site is important for maintaining stability by preserving the dimeric or tetrameric state, and is therefore critical for enzyme activity[6,7].

Since the resolution of the first crystal structure of G6PD was more than 20 years ago, it became evident that known point mutations proximal to the structural NADP+ site and dimer interface are associated with severe G6PD deficiency[8]. Further, recent work combining structural characterization and molecular dynamics simulation showed that the structural NADP+ site is not only crucial for dimer stabilization by ordering two beta-strands in the dimer interface, but that distant dimer-interface defects in dimerization-deficient mutants had long-range structural effects on the active site through additional interactions[9].

**Fig. 2 Structural stability and activity analysis of recombinant wild-type and Arg219Gly G6PD.** Thermal stability analysis of recombinant human (**A**) G6PD WT and (**B**) G6PD Meyer. Proteins were heated in the presence of 5x SYPRO Orange reporter dye and various NADP$^+$ concentrations (0, 10, and 100 μM). Protein unfolding was monitored by following the emission of fluorescence dye at 580 nm. Susceptibility to trypsin digestion of recombinant human (**C**) G6PD WT and (**D**) G6PD Meyer. Enzyme activity was measured after incubation with 0.5 mg/ml trypsin in the presence of various NADP$^+$ concentrations (0, 10, and 100 μM) at 25 °C for 5 min. Residual enzyme activity is expressed as a percentage of the activity for the same enzyme in the absence of trypsin. NT; no treatment. **E** Stability analysis of recombinant human G6PD variants upon Gdn-HCl treatment. Enzymatic activity was measured after incubation with various concentrations of Gdn-HCl for 2 h at 37 °C. $C_{1/2}$ is the Gdn-HCl concentration at which the enzyme loses 50% of its activity. **F** Thermal inactivation analysis of recombinant human G6PD variants. Enzyme activity was measured after the protein was heated for 20 min. $T_{1/2}$ is the temperature at which the enzyme loses 50% of its activity. Experiments were performed in triplicate, and data are presented as mean ± SD.

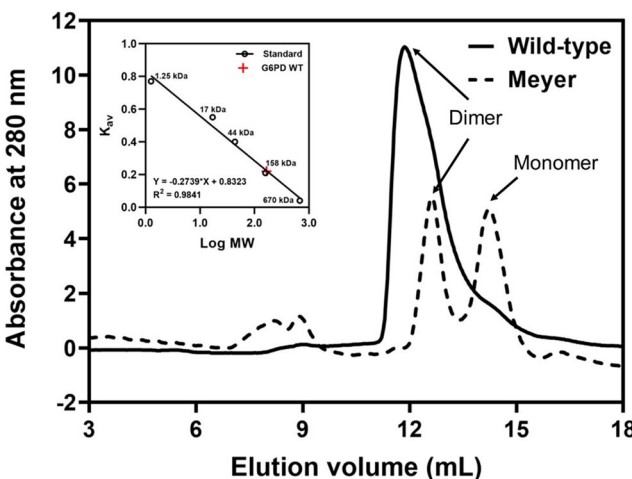

**Fig. 3 Size exclusion FPLC elution profiles of purified recombinant wild-type and Arg219Gly G6PD.** Proteins were loaded onto Superdex 200 Increase 10/300 equilibrated with 50 mM Tris-HCl and 150 mM NaCl.

It is, therefore, not surprising that Class A G6PD variants are those with mutations near the dinucleotide (NADP$^+$) binding site or the dimer interface, disturbing the oligomeric integrity, and hence enzyme stability and activity. Well-characterized examples are Palermo (Arg257Met), Bangkok (Lys275Asn), Hamburg (Pro276Leu), Bangkok noi (Phe501Cys), Wisconsin (Arg393Gly) and Nashville (Arg393His)[10,11].

It is interesting to note that in the Sao Paulo variant, isoleucine at position 220 is changed to methionine. This is directly adjacent to arginine 219, which in our variant is mutated to glycine. G6PD Sao Paulo, a normal activity variant, is classified as Class C (previously Class IV). The conservative isoleucine to methionine substitution preserves the hydrophobic nature of the branch chain and does not affect dimer stability[12]. This is not the case in our variant, which features a non-conservative arginine to glycine substitution at the dimer interface.

Through our comprehensive characterization of G6PD Meyer, we provide structural and biochemical evidence for the molecular basis of its pathogenicity. We show that arginine 219 substitution by glycine, previously uncharacterized, compromises dimerization, stability, and most importantly catalytic activity, explaining the phenotype of chronic hemolysis observed in our patient. Our work confirms the pathogenicity of this variant by independent functional assays, according to the criteria of the American College of Medical Genetics and Genomics[13].

It will be interesting to develop small molecule activators, along the lines of the well-characterized AG1 that promotes G6PD oligomerization by bridging the dimer interface at the structural NADP$^+$ binding sites of two monomers[6]. Indeed, oxidative damage is the trigger of hemolysis in G6PD deficiency. While its causes can be avoided by prevention measures, such as avoidance of fava beans and certain drugs, other causes are not as simple to prevent, including but not limited to viral infections, as described in our patient's medical history. Small molecule compounds that restore defective activity in Class A G6PD variants are therefore subjected of major current pharmacological efforts[14]. Our findings bear important implications in the path for such drug development, with the goal of rescuing the phenotype observed in G6PD Meyer and possibly other related variants. Small molecule activators could perhaps become a first-line treatment for G6PD deficiency in the future, in addition to, or in replacement of, prevention measures. Further, this strategy may be transposed to other enzymopathies where oligomerization is critical for enzyme function.

## Methods

**Exome sequencing.** The patient was a 15-year-old boy. Written informed consent from his mother was obtained prior to molecular studies. Whole-exome sequencing was performed on DNA extracted from venous blood, using the Twist Human Core Exome capture kit (Illumina NextSeq500 sequencer), followed by targeted analysis of the *G6PD* gene. Reads mapping and variant calling were performed using BWA 0.7.13, Picard 2.9.0 and GATK HaplotypeCaller 3.7 and annotated with annovar 2017-07-17 and UCSC RefSeq (refGene) downloaded on 2018-08-10. The variants were searched for in various databases including Genome Aggregation Database, ClinVar, Leiden Open Variation Database and Human Gene Mutation Database. Pathogenicity prediction scores were obtained for missense variants using SIFT, PolyPhen, MutationTaster and CADD. Polymerase chain reaction and Sanger sequencing for confirmation and familial segregation of the identified variant were done on saliva-extracted DNA.

Thorough biochemical and structural analysis was performed to understand the molecular mechanism underlying the enzyme deficiency causing the observed clinical manifestations.

**Construction of recombinant DNA.** The mutation (A655G) was created by sited-directed mutagenesis using pET28a-*G6PD* WT as a template[15]. Forward primer 5′ATTTGCCAACGGGATCTTCG 3′ and reverse primer 3′CGAAGATCCCGTTGGCAAAT 5′ were used. The polymerase chain reaction contained 1× Phusion HF buffer, 200 μM dNTPs, 0.5 μM of forward and reverse primers, 3% DMSO, 1 U of Phusion DNA polymerase (New England Biolabs) and 100 ng template DNA. The cycling parameters for site-directed mutagenesis were as follows: initial denaturation at 98 °C for 30 s, followed by 25 cycles of denaturation at 98 °C for 10 s, annealing at 50 °C for 30 s and extension at 68 °C for 3 min and 30 s. The polymerase chain reaction products were subjected to *Dpn*I digestion to remove the parental DNA for 2 h at 37 °C, followed by transformation into DH5α cells. The presence of desired mutation was confirmed by bi-directional DNA sequencing.

**Protein expression and purification**. G6PD protein was expressed in bacteria *E. coli* BL21 (DE3). A single colony of BL21 (DE3) carrying the desired recombinant plasmid was cultured overnight in Luria-Bertani medium containing 50 µg/ml kanamycin at 37 °C with 250 rpm shaking. The overnight cultures were inoculated into 1 L of Luria-Bertani medium containing 50 µg/ml kanamycin at a dilution of 1:100 and were grown at 37 °C with 250 rpm shaking until optical density at 600 nm reached 0.8–1.0. G6PD expression was then induced using 1 mM of isopropyl β-D-thiogalactoside (Merck) and cultured at 20 °C with 200 rpm shaking for 20 h before being harvested by centrifugation.

Cell pellets were resuspended in 20 mM sodium phosphate buffer pH 7.4 containing 300 mM NaCl and 10 mM imidazole and then subjected to sonication. Supernatant was collected after centrifugation for 60 min at $20,000 \times g$ and G6PD protein was purified using immobilized metal affinity chromatography (TALON resin, Takara bio). The unbound proteins were removed using 20 mM sodium phosphate buffer pH 7.4 containing 300 mM NaCl and 20 mM imidazole. After that, elution of G6PD protein was carried out using 20 mM sodium phosphate buffer pH 7.4 containing 300 mM NaCl and 40–400 mM imidazole. Imidazole was removed from protein solution by dialysis against 20 mM Tris-HCl pH 7.5 containing 10% glycerol at 4 °C for overnight. The purity of recombinant G6PD protein was analyzed by 12% sodium dodecyl-sulfate polyacrylamide gel electrophoresis. Protein concentration was determined by Bradford assay. The purified protein was stored at −20 °C until use.

**Determination of steady-state kinetic parameters**. Steady-state kinetic parameters were determined to assess the effects of Arg219Gly mutation on catalytic activity of G6PD enzyme. The standard reaction mixture contained 20 mM Tris-HCl pH 8.0, 10 mM MgCl₂, 500 µM G6P and 100 µM NADP⁺. The enzymatic reaction was initiated with the addition of G6PD enzyme and was monitored by following the formation of NADPH at 340 nm for 30 s using ultraviolet-visible spectrophotometer (Shimadzu). The steady-state kinetic parameters were determined by varying concentrations of one substrate (5–1000 µM G6P and 2.5–500 µM for NADP⁺) and fixing the concentration of the second substrate (500 µM for G6P and 100 µM for NADP⁺). The experiments were performed in triplicate. Kinetic parameters ($k_{cat}$, $K_m$ and $V_{max}$) were obtained from Michaelis-Menten plot.

**Secondary structure analysis**. To assess the effect of mutation on secondary structure of G6PD proteins, the secondary structure of the G6PD proteins was analyzed using CD. Far ultraviolet spectra (190–260 nm) were recorded using a Jasco spectrometer, model J-815, equipped with a Peltier temperature controller system in a 1 mm path-length quartz cuvette. For each sample, five scans were averaged, and the results of the buffer scans were subtracted.

**Structural stability analysis**. Several approaches were used to assess the effects of the mutation on protein structural stability. Initially, thermal stability and protein unfolding were determined in a 20 µL reaction, containing protein at a concentration of 0.25 mg/ml mixed with 5 × SYPRO Orange Protein Gel Stain (Thermo Fisher Scientific). The reaction mixtures were heated in a Light-Cycler 480 real-time polymerase chain reaction machine (Roche) at temperatures ranging from 20 to 80 °C, with excitation and emission wavelengths of 465 and 580 nm, respectively. The stabilizing effect of NADP⁺ was also investigated by incubating the protein with different concentrations of NADP⁺ (0, 10, and 100 µM). The melting temperature ($T_m$) of each G6PD variant was determined and defined as the temperature at which half of the protein unfolded.

Susceptibility of the Meyer variant to chemical denaturation, Gdn-HCl, was carried out. Upon exposure to Gdn-HCl, protein structure is perturbed and structural stability can be assessed by measuring residual enzyme activity. The protein (0.2 mg/ml) was treated with different concentrations of Gdn-HCl (0 to 0.5 M) for 2 h at 37 °C and residual enzyme activity was measured and expressed as a percentage of the same enzyme incubated without Gdn-HCl. $C_{1/2}$ was defined as the Gdn-HCl concentration at which the enzyme loses 50% of its activity.

To determine susceptibility of G6PD proteins to trypsin digestion, the protein (0.2 mg/ml) was treated with trypsin (0.5 mg/mL) for 5 min at 25 °C in the presence of different concentrations of NADP⁺ (0, 10 and 100 µM). The residual enzyme activity was measured and expressed as a percentage of the activity of the same enzyme incubated without trypsin.

Finally, thermal inactivation analysis was carried out. G6PD proteins were heated at different temperatures (25–65 °C) in a thermocycler for 20 min and then cooled to 4 °C. The residual enzyme activity was measured and expressed as a percentage of the activity of the same enzyme incubated at 25 °C. $T_{1/2}$ was defined as the temperature at which the enzyme loses 50% of its activity.

**Size exclusion chromatography**. G6PD enzyme is active as a dimer or tetramer; size exclusion chromatography was carried out to determine the effect of the mutation on the oligomeric state of the protein using AKTA fast protein liquid chromatography (FPLC, GE Healthcare) equipped with the Superdex 200 Increase 10/300 column. The recombinant protein (100 µg) was loaded onto the pre-equilibrated column and chromatography was carried out at a flow rate of 0.5 ml/min using 50 mM Tris-HCl pH 7.5 and 150 mM NaCl. The column was calibrated with Gel Filtration Standard (Biorad).

## Data availability

All raw data used to generate the figures are available in the Supplementary data file. The uncropped gel scan is clearly labeled and matches the gel presented in Supplementary Fig. 1. The G6PD R219G variant has been deposited in ClinVar under the accession number SCV004101564.

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

## Acknowledgements

We thank Prof. Ronen Marmorstein (University of Pennsylvania) for his valuable input on the structural prediction of G6PD Meyer's functionality. This research was supported by Mahidol University (Fundamental Fund: fiscal year 2023 by National Science Research and Innovation Fund (NSRF)) to UB. The funder had no role in study design, data collection and analysis, decision to publish, or preparation of the manuscript.

## Author contributions

Conceptualization, methodology, project administration, resources, writing—original draft: O.Z. and U.B.; Data curation: O.Z., U.B., T.N., and J.-L.B.; Clinical analysis: O.Z., L.C., and L.Q.; Formal analysis: O.Z., K.C., and U.B.; Funding acquisition: U.B.; Investigation: O.Z., K.C., and U.B.; Validation: O.Z., U.B., L.C., L.Q., T.N., J.-L.B.; Visualization: O.Z., K.C., U.B.; Writing—review & editing: O.Z., K.C., L.C., L.Q., T.N., J.-L.B., and U.B.

## Competing interests

The authors declare no competing interests.

## Ethics approval and consent to participate

As the patient was under the age of 16, written informed consent to participate was obtained from his mother.

## Informed consent

Written informed consent for publication of clinical details was obtained from the patient's mother, available on request.

## Additional information

**Consent to publish** Written informed consent for publication of clinical details was obtained from the patient's mother, available on request.

