## [Peer Review File · Communications Biology]

Reviewers' comments:

Reviewer #1 (Remarks to the Author):

The overall composition of the manuscript is good. The paper is scientifically and methodologically accurate. This manuscript find interest many readers. However, my recommendation is 'Minor Revision'. More detailed comments are given below.

1) The main problem is that the authors must justify what is the advantage of their study with respect to previous studies.

The authors mention that "Arginine 219 by Glycine named G6PD Geneva, is a novel Class A mutant. However, this mutant was previously reported by Citana et al. (2014) as G6PD Meyer and included in the List of G6PD mutations performed in a review by Luzzato et al. (2020). Please verify and cite these two previously works. Besides, modified the title and the main document.

Citana A, Leone D, Moruzzi F, et al. G6PD-Meyer: a new mutation causing compensated chronic haemolysis. European Human Genetics Conference. 2014

Luzzatto L, Ally M, Notaro R. Glucose-6-phosphate dehydrogenase deficiency. Blood. 2020 Sep 10;136(11):1225-1240. doi: 10.1182/blood.2019000944.

2) Introduction section. The authors need to reinforce their focus on why they carry out this work.

3) Before the Materials and Methods section, the authors must indicate the techniques used to respond to the question of this study.

4) Results section: I suggest that the results be separated into subsections as presented in materials and methods.

5) Results section: A short introduction and conclusion in each section must be provided before describing all the experiments and results that support each section. It is to guide the reader to understand the importance of the results.

6) Lines 137-138. The authors mention that "to assess the effects of A655G mutation on catalytic activity of G6PD enzyme" Why did they mention A655G when they are evaluating the effect of amino acid change in the protein. Please verify.

7) Line 211. Nitrogen eta 1? I the same for line 213 Nitrogen eta 2?

8) Table 1. Reduce the at one digit the values. Is not true that exact values with two digits were obtained.

9) Recommend including a table with the data of the patients, age, and gender.

10) A conclusion section is necessary in the main document.

Reviewer #2 (Remarks to the Author):

The authors report a new G6PD variant (they named Geneva) found in a 15-year-old man known for G6PD deficiency with undetectable activity (Class A). This variant consists of the substitution of arginine 219 for glycine. Through a well-conducted comparative study of wild-type and mutant G6PD recombinant proteins, they provided structural and biochemical evidence for the molecular basis of Geneva pathogenicity. From the obtained results, they proposed a critical role for arginine 219 in G6PD dimer formation. The oligomeric integrity (dimer or tetramer) of G6PD is essential for enzyme stability and activity.

The introduction provides sufficient background information about the subject. Results are clearly and logically presented and figures/table support the text. Discussion is appropriate and conclusions are based on results. I consider that the findings of this work will be of interest to others in the community and the wider field.

Nonetheless, I found some points that should be addressed:

1. Although the authors do not mention if the recombinant proteins were cloned in the pET28a plasmid

fused to a His(6)-tag, I suppose that it is the case since the proteins were purified using immobilized metal affinity chromatography (TALON resin). However, the purification procedure does not indicate an enzymatic digestion step for the His-tag removal. The authors should justify why they consider this step unnecessary when the study is based on structural and enzymatic characterization of wild-type and mutant G6PD and if the proteins contain a His-tag, they are not in their native form.

2. In the Methods section:

Lane 131: "The purity of recombinant G6PD protein was analyzed by 12% SDS-PAGE. Protein concentration was determined by Bradford assay". The authors should indicate the purity grade of the wild-type and mutant G6PD obtained since the Bradford assay determines the total proteins in a sample. Maybe, a supplementary figure showing the SDS-PAGE could be included.

3. Use consistently NADP⁺ throughout the text. In some sentences, it appears as NADP (e.g. lanes 267, 270, 274, etc)

4. Lane 239: "Fluorescence-based thermal shift assays revealed that the T_m of the mutant (52.99 °C) was ~6 °C lower than the WT enzyme (47.04 °C)". The T_m values are inverted: T_m of the mutant is 47.04 °C and the T_m of WT enzyme is 52.99 °C. Please correct.

5. Lane 287: "It interesting to note...". In this sentence, the verb to be is missing. Please correct.

6. Figure 5: If "NT" bars refer to 100% residual activity I consider that it is not necessary to include the error bars in them.

7. Figure Legends.

Figure 2. Lane 400: "NAPD" should be replaced by NADP⁺

Reviewer #3 (Remarks to the Author):

In the presented manuscript, the authors detail the case of a 15-year-old male with a novel G6PD variant, specifically the Arg219Gly mutation. The study aims to elucidate the molecular consequences of this new G6PD variant. Through a series of biochemical assays, the authors convincingly demonstrate that this mutation hampers both the activity and stability of the G6PD dimers. Their data suggests that this mutation disrupts the dimerization interface of G6PD, compromising its thermostability. While the manuscript is cogently written and offers valuable insights, certain aspects of the methods section need elaboration, such as specifics on reagents used and software/methods employed for calculating kinetic parameters.

I have a few queries and recommendations for the authors:

In the size exclusion chromatography data, the WT protein seems to lack a tetrameric form, while the mutated version displays two apparent peaks that might indicate tetramers. Could the authors provide clarity on this?

Purity verification through SDS-PAGE is mentioned, but no comments on the degree of purity or related images are provided.

Details like the duration for which NADPH absorbance was measured in activity assays and protein concentrations in both the thermal and trypsin-mediated inactivation assays are missing. Such critical information needs inclusion in the methods section.

Minor suggestions:

In Figure 4, panels C and D, it would be helpful if the symbols for Wild type and A655G were distinct for clarity.

Figure 2 would be enhanced with an additional model that highlights the glycine substitution and elucidates the affected interactions.

To streamline the presentation, Figures 1, 2, and 3 can be integrated, and Figures 4 and 5 merged into a single composite figure.

Overall, the study brings forth intriguing findings, and with these suggested revisions, it can provide a clearer, more comprehensive understanding to its readers.

Response to reviewers' comments

Reviewers' comments:

Reviewer #1 (Remarks to the Author):

The overall composition of the manuscript is good. The paper is scientifically and methodologically accurate. This manuscript find interest many readers. However, my recommendation is 'Minor Revision'. More detailed comments are given below.

1) The main problem is that the authors must justify what is the advantage of their study with respect to previous studies.

The authors mention that "Arginine 219 by Glycine named G6PD Geneva, is a novel Class A mutant. However, this mutant was previously reported by Citana et al. (2014) as G6PD Meyer and included in the List of G6PD mutations performed in a review by Luzzato et al. (2020). Please verify and cite these two previously works. Besides, modified the title and the main document. Citana A, Leone D, Moruzzi F, et al. G6PD-Meyer: a new mutation causing compensated chronic haemolysis. European Human Genetics Conference. 2014 Luzzatto L, Ally M, Notaro R. Glucose-6-phosphate dehydrogenase deficiency. Blood. 2020 Sep 10;136(11):1225-1240. doi: 10.1182/blood.2019000944.

Response: The title and content have been modified accordingly. Indeed, the mutant has been previously reported in a poster abstract at the 2014 European Genetics Conference. This poster was cited in the supplementary references of the mentioned review. We have included this additional poster reference in our manuscript. The review already was.

Our study provides detailed clinical information and molecular characterization of the mutant, which has never been characterized (or formally published) previously (a poster presentation is not a publication). While red cell G6PD activity assays do indeed provide information on enzyme activity, independent functional assays that control for all variables and only analyze the effect of a mutation are essential for extensive characterization. This is in line with pathogenicity criteria of the American College of Genetics and Genomics, which we refer to at the end of the manuscript.

2) Introduction section. The authors need to reinforce their focus on why they carry out this work.

Response: The introduction has been amended accordingly (see Response to Question 1 and modified Introduction)

3) Before the Materials and Methods section, the authors must indicate the techniques used to respond to the question of this study.

Response: In the introduction, before Materials and Methods, we indicate the background and reason for carrying out the study. The techniques used are fully indicated in the Materials and Methods section, as is customary.

4) Results section: I suggest that the results be separated into subsections as presented in materials and methods.

Response: In the revised manuscript, the results were separated into subsections as suggested.

5) Results section: A short introduction and conclusion in each section must be provided before describing all the experiments and results that support each section. It is to guide the reader to understand the importance of the results.

Response: The results section was revised.

6) Lines 137-138. The authors mention that "to assess the effects of A655G mutation on catalytic activity of G6PD enzyme" Why did they mention A655G when they are evaluating the effect of amino acid change in the protein. Please verify.

Response: The variant name was changed to Arg219Gly.

7) Line 211. Nitrogen eta 1? | the same for line 213 Nitrogen eta 2?

Response: Arginine has one nitrogen epsilon and two nitrogen eta. Choosing which is eta1 and eta 2 is arbitrary. What is essential is the description of the stabilizing interactions with these nitrogens, which we have addressed.

8) Table 1. Reduce the at one digit the values. Is not true that exact values with two digits were obtained.

Response: Values in Table 1 was reduced to one digit as suggested.

9) Recommend including a table with the data of the patients, age, and gender.

Response: We don't see a real interest in including an additional table with this information. This would have little added value in a manuscript with already many figures. To streamline the presentation, as suggested by one reviewer, we have in fact removed the family tree figure (previously Figure 1) and merged Figures 2 and 3 into one figure (Figure 1).

10) A conclusion section is necessary in the main document.

Response: A conclusion section has been added, which summarizes the findings and provides future outlook comments.

Reviewer #2 (Remarks to the Author):

The authors report a new G6PD variant (they named Geneva) found in a 15-year-old man known for G6PD deficiency with undetectable activity (Class A). This variant consists of the substitution of arginine 219 for glycine. Through a well-conducted comparative study of wild-type and mutant G6PD recombinant proteins, they provided structural and biochemical evidence for the molecular basis of Geneva pathogenicity. From the obtained results, they proposed a critical role for arginine 219 in G6PD dimer formation. The oligomeric integrity (dimer or tetramer) of G6PD is essential for enzyme stability and activity.

The introduction provides sufficient background information about the subject. Results are clearly and logically presented and figures/table support the text. Discussion is appropriate and conclusions are based on results. I consider that the findings of this work will be of interest to others in the community and the wider field.

Nonetheless, I found some points that should be addressed:

1. Although the authors do not mention if the recombinant proteins were cloned in the pET28a plasmid fused to a His(6)-tag, I suppose that it is the case since the proteins were purified using immobilized metal affinity chromatography (TALON resin). However, the purification procedure does not indicate an enzymatic digestion step for the His-tag removal. The authors should justify why they consider this step unnecessary when the study is based on structural and enzymatic characterization of wild-type and mutant G6PD and if the proteins contain a His-tag, they are not in their native form.

Response: Yes, the *G6PD* gene was cloned with His(6)-tagged to facilitate the following purification step. The His-tag was not removed due to the fact that it does not interfere with enzyme activity, structure or protein stability (S. Gomez-Manzo et al., *Protein J.* 32 (2013) 585–592.). The information has been added in the revised manuscript.

2. In the Methods section:

Lane 131: "The purity of recombinant G6PD protein was analyzed by 12% SDS-PAGE. Protein concentration was determined by Bradford assay". The authors should indicate the purity grade of the wild-type and mutant G6PD obtained since the Bradford assay determines the total proteins in a sample. Maybe, a supplementary figure showing the SDS-PAGE could be included.

Response: The information on the degree of purity of has been added in the revised manuscript. The SDS-PAGE was included as a supplementary figure 1.

3. Use consistently NADP⁺ throughout the text. In some sentences, it appears as NADP (e.g. lanes 267, 270, 274, etc)

Response: The term was revised as suggested.

4. Lane 239: "Fluorescence-based thermal shift assays revealed that the T_m of the mutant (52.99 °C) was ~6 °C lower than the WT enzyme (47.04 °C)". The T_m values are inverted: T_m of the mutant is 47.04 °C and the T_m of WT enzyme is 52.99 °C. Please correct.

Response: The T_m values for WT and mutant were corrected.

5. Lane 287: "It interesting to note...". In this sentence, the verb to be is missing. Please correct.

Response: The sentence has been revised.

6. Figure 5: If "NT" bars refer to 100% residual activity I consider that it is not necessary to include the error bars in them.

Response: Error bars have been removed from NT.

7. Figure Legends.

Figure 2. Lane 400: "NAPD" should be replaced by NADP⁺

Response: The term was revised as suggested.

Reviewer #3 (Remarks to the Author):

In the presented manuscript, the authors detail the case of a 15-year-old male with a novel G6PD variant, specifically the Arg219Gly mutation. The study aims to elucidate the molecular consequences of this new G6PD variant. Through a series of biochemical assays, the authors convincingly demonstrate that this mutation hampers both the activity and stability of the G6PD dimers. Their data suggests that this mutation disrupts the dimerization interface of G6PD, compromising its thermostability. While the manuscript is cogently written and offers valuable insights, certain aspects of the methods section need elaboration, such as specifics on reagents used and software/methods employed for calculating kinetic parameters.

I have a few queries and recommendations for the authors:

In the size exclusion chromatography data, the WT protein seems to lack a tetrameric form, while the mutated version displays two apparent peaks that might indicate tetramers. Could the authors provide clarity on this?

Response: The figure was revised to indicate monomeric and dimeric forms of the recombinant proteins. Two small peaks at elution volume around 7-9 ml were not the tetrameric form of the mutant protein. According to the calibration curve, the tetrameric form should be eluted out at the elution volume of around 11.20 ml. Those two peaks are rather the aggregated form of the mutant protein as it was found that the void volume for size exclusion chromatography was at elution volume of 8.38 ml.

Purity verification through SDS-PAGE is mentioned, but no comments on the degree of purity or related images are provided.

Response: The information on the degree of purity of has been added in the revised manuscript. The SDS-PAGE was included as a supplementary figure 1.

Details like the duration for which NADPH absorbance was measured in activity assays and protein concentrations in both the thermal and trypsin-mediated inactivation assays are missing. Such critical information needs inclusion in the methods section.

Response: The information has been added in the revised manuscript.

Minor suggestions:

In Figure 4, panels C and D, it would be helpful if the symbols for Wild type and A655G were distinct for clarity.

Response: For clarity, the figures have been revised to make the symbols more distinct.

Figure 2 would be enhanced with an additional model that highlights the glycine substitution and elucidates the affected interactions.

Response: Given that the interactions mediated by arginine 219 are well described in the text in reference to Figure 1B, adding an additional figure with modeled glycine 219 would have little added value in our view. Not only would this be a model and not an actual structure, but also it is quite intuitive to imagine that with only a hydrogen side chain instead of arginine, all the described interactions with arginine would be absent.

To streamline the presentation, Figures 1, 2, and 3 can be integrated, and Figures 4 and 5 merged into a single composite figure.

Response: Figure 1 (family tree) has been removed as it has little added value. Figures 2 and 3 have been integrated as Figure 1 while Figures 4 and 5 were merged as Figure 2 in the revised manuscript.

Overall, the study brings forth intriguing findings, and with these suggested revisions, it can provide a clearer, more comprehensive understanding to its readers.

Response: We thank the reviewer for constructive comments and thorough review, which helped improve the manuscript.

REVIEWERS' COMMENTS:

Reviewer #2 (Remarks to the Author):

I consider that the authors have addressed all the points the Reviewers raised and implemented the appropriate changes.

The new version of the manuscript contains all the reviewer's recommendations, providing readers with a clearer, more comprehensive understanding.

Reviewer #3 (Remarks to the Author):

the manuscript has been improved substantially. I have no further questions.